# Accuracy of Endodontic Access Cavities Performed Using an Augmented Reality Appliance: An In Vitro Study

**DOI:** 10.3390/ijerph191811167

**Published:** 2022-09-06

**Authors:** Vicente Faus-Matoses, Vicente Faus-Llácer, Tanaz Moradian, Elena Riad Deglow, Celia Ruiz-Sánchez, Nirmine Hamoud-Kharrat, Álvaro Zubizarreta-Macho, Ignacio Faus-Matoses

**Affiliations:** 1Department of Stomatology, Faculty of Medicine and Dentistry, University of Valencia, 46010 Valencia, Spain; 2Department of Implant Surgery, Faculty of Health Sciences, Alfonso X El Sabio University, 28691 Madrid, Spain; 3Department of Surgery, Faculty of Medicine and Dentistry, University of Salamanca, 37008 Salamanca, Spain

**Keywords:** augmented reality, computer-assisted treatment, endodontic access cavity, image-guided treatment, navigation system, real-time tracking

## Abstract

Introduction: The purpose of this study was to compare and contrast the accuracy of endodontic access cavities created using an augmented reality appliance to those performed using the conventional technique. Materials and Methods: 60 single-rooted anterior teeth were chosen for study and randomly divided between two study groups: Group A—endodontic access cavities created using an augmented reality appliance as a guide (*n* = 30) (AR); and Group B—endodontic access cavities performed with the manual (freehand) technique (*n* = 30) (MN). A 3D implant planning software was used to plan the endodontic access cavities for the AR group, with a cone-beam computed tomography (CBCT) and 3D intraoral surface scan taken preoperatively and subsequently transferred to the augmented reality device. A second CBCT scan was taken after performing the endodontic access cavities to compare the planned and performed endodontic access for accuracy. Therapeutic planning software and Student’s *t*-test were used to analyze the cavities at the apical, coronal, and angular levels. The repeatability and reproducibility of the digital measurement technique were analyzed using Gage R&R statistical analysis. Results: The paired *t*-test found statistically significant differences between the study groups at the coronal (*p* = 0.0029) and apical (*p* = 0.0063) levels; no statistically significant differences were found between the AR and MN groups at the angular (*p* = 0.6596) level. Conclusions: Augmented reality devices enable the safer and more accurate performance of endodontic access cavities when compared with the conventional freehand technique.

## 1. Introduction

The preparation of endodontic access cavities is considered the first step in achieving a successful endodontic treatment outcome, but it is also one of the most frustrating and challenging aspects. Therefore, the access cavity’s proper design and adequate preparation are crucial for ensuring quality endodontic treatment, preventing iatrogenic problems, and avoiding endodontic failure [1]. It is the first invasive step of any root canal treatment and therefore has a significant impact on the stability and outcome of results, as well as tooth longevity [2]. Furthermore, a lack of accuracy can make it more challenging to locate the root canals and may result in unexpected complications [3]. Bacteria are the main cause of periapical and pulpal inflammation, and failure to adequately eradicate bacteria and their by-products may lead to persistent irritation and impede the healing process [4]; therefore, many authors have highlighted the relevance of using a sodium hypochlorite disinfection agent in removing bacteria inside the root canal system. Moreover, Iandolo et al. reported that heating sodium hypochlorite increases the antimicrobial activity of the irrigating agent [5]. Additionally, missed root canals can harbor a number of microorganisms, which are one of the primary causes of persistent apical periodontitis and can negatively impact treatment outcomes [6]; therefore, it is crucial that all root canals be located and properly disinfected.

Dynamic guidance involves using computer-aided surgical techniques analogous to the technology used for satellite navigation or global positioning systems [7]. Computer-aided dynamic navigation systems provide a direct view of the surgical field and enable endodontists to relocalize the position of the endodontic access cavity [8]. Furthermore, the use of dynamic navigation systems provides several advantages over traditional static guided surgery, including reduced costs and time due to the lack of need for impression and laboratory procedures required when using a static guiding system. Another benefit of dynamic guided systems is the direct view of the surgical field they provide and the possibility of using standard drills, which are optimal in cases of reduced mouth opening [9].

Additionally, endodontic access cavities have also been performed using virtual reality [10]. Augmented reality (AR) techniques entail the co-virtualization of a real-time image and a virtual image, enabling the user to simultaneously interact with and observe the components of both images [11]. AR can be viewed on a traditional monitor or when using a wearable system directly in the visual field of the surgeon [12]. While both artificial intelligence (AI) and intelligence augmentation (IA) use the same basic technology, IA works by placing humans at the focal point, with machines serving them to achieve the objective, while AI places technology first and tends to result in independent machines being manufactured [13].

A similar methodology was previously used by Fraguas de San José et al. to analyze the wear of dental implant drills after clinical use [14], Alakabani et al. to analyze the removal capability of carrier-based root canal filling material from straight root canal systems [15], and Faus-Matoses et al. to analyze the wear of CM-Wire NiTi alloy endodontic reciprocating files after root canal treatment [16].

The purpose of the present study was to compare and contrast the accuracy of endodontic access cavities created using an augmented reality appliance compared with those performed using the conventional technique. The null hypothesis (H_0_) assumes that there is no difference in the accuracy of endodontic access cavities between those performed using an augmented reality appliance versus those performed with the conventional technique.

## 2. Materials and Methods

### 2.1. Study Design

A total of 60 single-rooted anterior teeth (lower central incisors) were selected for study from January to March 2022 at the Dental Center of Innovation and Advanced Specialties at Alfonso X El Sabio University in Madrid, Spain. The teeth all presented without any caries or restorations, and they had been extracted for periodontal reasons. The clinical crown dimensions of the teeth ranged from 8.5 to 9.5 mm in height, 4.5 to 5.5 mm in M–D width, and 5.5 to 6.5 mm in V-L/P width. Researchers carried out a randomized controlled experimental trial in conformance with the principles outlined by the German Ethics Committee in its statement on the usage of organic tissues for medical research (Zentrale Ethikkommission, 2003). The Alfonso X El Sabio University Ethics Committee approved the study (process no. 03/2020). All patients were informed of the study and consented to their teeth being transferred.

### 2.2. Experimental Procedure

The teeth were randomly divided (Epidat 4.1, Galicia, Spain) into one of two groups: Group A: guided technique using an augmented reality appliance (Hololens2, Redmond, WA, USA) to perform endodontic access cavities (*n* = 30) (AR); and Group B: a manual (freehand) technique to perform endodontic access cavities (*n* = 30) (MN). The teeth were placed into two epoxy resin models (ref. 20-8130-128, EpoxiCure^®^, Buehler, IL, USA), with 10 teeth in each model. A power of 80.00% was calculated using a bilateral Student’s *t*-test of the two independent samples; this was used to analyze the variation from the null hypothesis H_0_: μ_1_ = μ_2_, bearing in mind that with a significance level of 5.00%, more than 20 teeth must be included.

A preoperative CBCT scan (WhiteFox, Acteón Médico-Dental Ibérica S.A.U.-Satelec, Merignac, France) was taken of the epoxy resin model belonging to the AR study group using the following exposure parameters: 105.0 kV peak, 7.20 s, 8.0 mA, and a field of view of 15 × 13 mm. A 3D surface scan was subsequently performed in conjunction with a 3D intraoral scan (True Definition, 3M ESPE ™, Saint Paul, MN, USA) using 3D in-motion video imaging technology to create an accurate digital file in the “Standard Tessellation Language” (STL) file format. The datasets taken from this digital workflow were imported into a 3D implant planning software (NemoScan^®^, Nemotec, Madrid, Spain) to plan the virtual endodontic access cavities by overlaying the CBCT data over the 3D surface scan to align the key points of the teeth crowns. The implant planning software was also used to design a virtual implant bur to perform an access cavity inside each tooth, with a 1.2 mm diameter, 14 mm total length, and 11 mm drilling depth to provide direct access to the root canal system (Figure 1).

After designing the endodontic access cavities, the STL digital file of the endodontic access cavities was imported into a mixed reality appliance (Hololens2, Redmond, WA, USA) to enable the visualization of the endodontic access cavities (Figure 2).

Researchers used a diamond bur with a 1.2 mm diameter on the active part, 14 mm total length, and 11 mm working length (Ref. 882 314 012, Komet Medical, Lemgo, Germany).

A preoperative CBCT scan was taken of the epoxy resin model belonging to the MN control group, and the datasets were then added to 3D implant planning software (NemoScan^®^, Nemotec, Madrid, Spain) for virtual planning of the straight access cavities; no templates were used. The same operator performed all endodontic access cavities per the technique suggested by Gilboe et al. [17] and Mauger et al. [18].

The root canal systems were explored in both study groups using a #10 K-file (Dentsply Maillefer, Ballaigues, Switzerland) to clinically confirm the canal locations.

### 2.3. Measurement Procedure

Postoperative CBCT scans were taken of both study groups after performing the access cavities. These scans were subsequently uploaded to the 3D implant planning software (NemoScan^®^, Nemotec, Madrid, Spain) along with the virtually planned access cavities. These were then overlaid and aligned to calculate the horizontal deviation (measured at the apical endpoint and coronal entry point) and deviation angle (measured in the center of the cylinder). The deviations were analyzed in the coronal, sagittal, and axial views (Figure 3).

### 2.4. Validation of the Repeatability and Reproducibility

To validate the repeatability of this new protocol, the measurements described above were calculated six times using the same operator (Operator A). The measurements were calculated six times by another operator (Operator B) to validate the reproducibility of this new measurement technique. The degree of agreement between the examiners was evaluated using the kappa index [19]. The proportion of correct answers observed ranged between 86% and 90%, and the values of the kappa test were between 0.61 and 0.69, which are considered substantial based on the Landis and Koch scale [20].

### 2.5. Statistical Analysis

The variables under study were all recorded using SPSS 22.00 statistical analysis software for Windows. Quantitative variables were expressed as means and standard deviation (SD) for the descriptive statistical analysis. The comparative analysis was conducted using Student’s *t*-test to compare the mean deviation between planned and performed endodontic access cavities; *p* < 0.05 was considered statistically significant as the variables were normally distributed. Gage R&R statistical analysis was conducted to analyze the repeatability and reproducibility of this measurement technique.

## 3. Results

Table 1 shows the mean values, SD values for deviation, and statistical significance of the recorded coronal deviation (mm), apical deviation (mm), and angular deviation (°).

The two sample *t*-tests found a statistically significant difference between the deviations in the coronal entry points of AR and MN (*p* < 0.001) (Table 1 and Figure 4A). The paired Student’s *t*-test found a statistically significant difference in apical deviations between the AR and MN groups (*p* < 0.001) (Table 1 and Figure 4B). On the other hand, the paired Student’s *t*-test found no statistically significant differences in the angular deviation between the AR and MN groups (*p* < 0.001) (Table 1 and Figure 4C).

The augmented reality appliance successfully located the root canal system for all of the endodontic access cavities performed, while those performed in the MN control group resulted in two missed root canals and one root perforation.

Table 2 and Figure 5 display the means and SD values necessary to analyze the repeatability of the measurement technique for the coronal entry point, apical endpoint, and angular measurements.

The Gage R&R statistical analysis of the measurement technique showed that the variabilities attributable to the repeatability of the measurement technique were 2.7%, and the variabilities attributable to the reproducibility of the measurement technique were 2% of the total variability of the samples. The technique demonstrated high repeatability and reproducibility for the measurement technique since the values of repeatability and reproducibility were under 10% (Figure 5).

## 4. Discussion

The present study’s findings reject the null hypothesis (H_0_) that there is no difference in the accuracy of endodontic access cavities between those performed using an augmented reality appliance and those performed using the conventional technique.

In recent years, computer-aided dynamic navigation techniques have been used for dental implant surgery [3,21,22,23,24]. These techniques are more predictable than conventional freehand techniques and have more favorable results, improving the accuracy of placement of dental implants and resulting in a reduction in clinical complications [21,22,23,24]. The present study is the first to test the use of a computer-aided dynamic navigation system in performing endodontic access cavities.

Computer-aided static navigation techniques using surgical templates were previously developed in the hopes of improving the accuracy of the placement of dental implants. These techniques use preoperative CBCT and 3D surface scans for surgical planning, enabling improved treatment planning and better comprehension of the individual anatomy of each specific case [25,26]. The digital workflow aids in collecting datasets, which are then imported into a 3D implant planning software. Here, virtual templates are designed by aligning key points of the teeth crowns to match the CBCT and 3D surface scan data [27]. Compared with the freehand dental implant placement technique, the static guidance technique has a reported mean angle deviation of 0.621°, mean coronal deviation of 0.193 mm, and mean apical deviation of 0.277 mm [27]. The technique has been used in endodontic treatment for improved conservative access cavities [2,28]. Zehnder et al. (2016) obtained a mean angle deviation of 1.81°, mean coronal deviation of 0.16–0.21 mm, and mean apical deviation of 0.17–0.47 mm in endodontic access cavities performed using an implant bur that was 1.5 mm in diameter [2]. The lack of accuracy of the static guidance technique may negatively impact the locations of root canals and lead to perforated roots or fragile teeth. Giacomino et al. (2018) found evidence of unlocated canals in 8.3% of cases, as well as an average substance loss of 9.8 mm [29]. For the present in vitro study, all planned root canals in the computer-aided navigation study group were located without complications. In the MN control group, however, two root canals were missed, and one root was perforated. The high angular deviation and narrow root anatomy of lower central incisors in the MN control group may have contributed to the appearance of root perforation and missed root canals. The apical and angular deviations are the most relevant parameters analyzed in this study, as the apical deviation impacts the risk of root perforation and missed root canals. It is directly correlated with the angular deviation in cases of calcific metamorphosis because the horizontal apical endpoint deviation increases with a high angular deviation. As it shows better results than the traditional manual technique, this lack of accuracy has promoted computer-aided dynamic navigation technologies for their potential application in the clinical transfer of positions of endodontic access cavities that have been planned virtually. Furthermore, one of the primary benefits of this technology is the ability to change the direction of the access cavity in real time. The stereoscopic motion-tracking optical cameras dynamically recognize and triangulate the optical reference markers, guiding the access cavity at the preoperatively planned depth, pathway, and angle. Additional benefits include greater visibility of the dental field during clinical procedures and greater preservation of tooth tissue, which reduces the risk of iatrogenic damage [30,31].

Computer-aided dynamic navigation can be particularly useful in cases of malformed dental development, such as dens invaginatus or evaginatus, in which several conservative accurate access cavities are needed for the localization of individual root canals [3,24], even when performing a conservative osteotomy and root-end section during endodontic microsurgery [32,33].

Static guidance techniques require that several surgical templates be developed to enable direct access to individual root canals in posterior teeth. This is not an issue with computer-aided dynamic navigation, as the required access cavities are planned before the procedure [27,34,35]. A computer-aided static navigation technique using surgical templates circumvents the need for drilling guidance during the surgical intervention [27,34,35,36]. As a result, the accuracy of endodontic cavity access directly depends on how the surgical template is designed and manufactured; any inaccuracies during the manufacturing process may increase the risk of intraoperative complications. In contrast, computer-aided dynamic navigation systems enable a direct view of the surgical field and enable the operator to relocalize the position of the endodontic access cavity. These systems are also particularly useful in cases of limited mouth opening, as well as treatments in the posterior region [27,35,36]. The primary drawback of computer-aided dynamic navigation systems is that it can be difficult to maintain the visibility of the system display during the procedure [7]. That being said, an augmented reality device can be used to project a virtual image of the system without losing visibility of the therapeutic field [27]. Preoperative planning information is displayed on the mobile unit on a mounted laptop computer. When the “target” is displayed on the laptop unit, the operator looks away from the tooth instead of directly at it. The ability to control the handpiece and maintain the drill entry point angle, depth, and pathway requires a certain level of technical skill, manual dexterity, and hand–eye coordination, with a steep learning curve [3,21,22,23,24,30,31]. Fahim et al. have also highlighted the impact of augmented reality and virtual reality on the patient’s experience since clinicians can use augmented reality and virtual reality technology to show their patients the expected outcomes before they undergo dental procedures. Additionally, augmented reality and virtual reality can be implemented to overcome dental phobia, which is commonly experienced by pediatric patients [37]. However, these researchers did not consider the application of augmented and virtual reality for learning or treatment.

In the present in vitro study, endodontic access cavities were performed more accurately in the AR study group than in the MN study group; however, these differences were not statistically significant. This could be due to the limited sample size, the learning curve necessary to use computer-aided dynamic navigation systems successfully, and the depth of the established endodontic access cavity; the angular deviations seen in the MN study group revealed differences from the AR study group close to statistical significance, signifying that if the endodontic access cavities were deeper, the horizontal deviations at the apical endpoint would increase between the two study groups. Additionally, no statistically significant differences were found in the fracture resistance of single-rooted teeth submitted to conservative endodontic access cavities compared to traditional endodontic accesses cavities. That being said, the average fracture resistance values of conservative endodontic access cavities were higher in absolute value than the average fracture resistance values of conventional endodontic accesses cavities. Furthermore, conservative endodontic access cavities enabled a lower canal wall area to remain untouched by the endodontic rotary instruments, with less dentin volume removed in absolute values [38], bearing in mind that the ability to clean and shape is the main indicator for a root canal treatment with a good prognosis [39]. The 30 conservative endodontic access cavities were performed manually, without computer-aided static or dynamic techniques, in accordance with the endodontic access cavities design described by Clark and Khademi in 2010. It is therefore challenging to ensure that conservative endodontic access cavities have the same size, design, and location, which is necessary to compare them.

The main strength of this study is the novel approach of augmented reality application for access cavities’ design. However, the experimental nature of this study avoids transferring the results to the clinical setting; therefore, further clinical studies are required to validate this technique for endodontic application.

This study found that endodontic access cavities performed using computer-aided static and dynamic navigation systems were more accurate than those performed using a manual (freehand) technique. Further research is needed to establish the accuracy of endodontic access cavities performed with new technologies, as well as their potential for clinical complications.

## 5. Conclusions

Bearing in mind the limitations of this study, the results indicate that augmented reality technology is a viable alternative for access cavities in endodontics; however, clinical studies are required to validate this procedure clinically.

## Figures and Tables

**Figure 1 ijerph-19-11167-f001:**
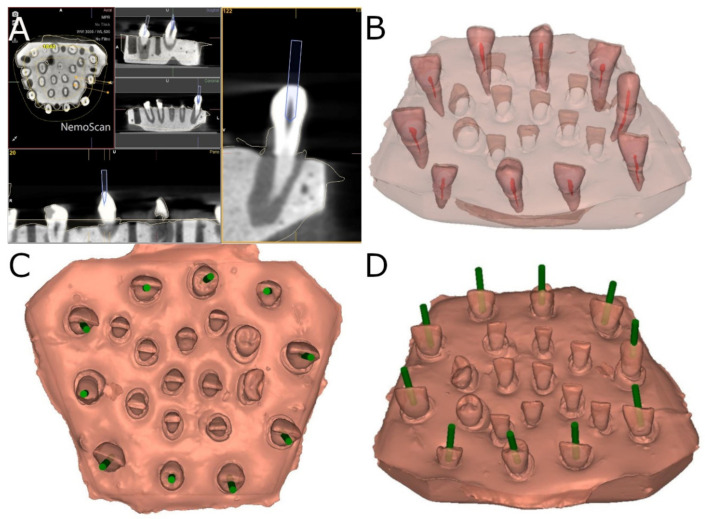
(**A**) Planned endodontic access cavities based on CBCT scans (**B**) after identifying the root canal system of the teeth. (**C**) Occlusal view and (**D**) oblique view of the planned endodontic access cavities (green cylinders).

**Figure 2 ijerph-19-11167-f002:**
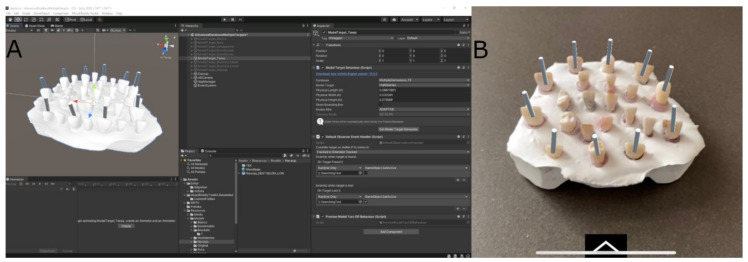
(**A**) Screenshot of augmented reality device software used for planning process and (**B**) image obtained with the augmented reality appliance with virtual endodontic access cavities (gray cylinders).

**Figure 3 ijerph-19-11167-f003:**
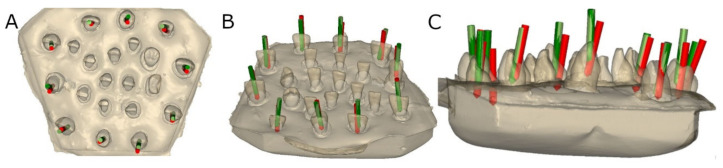
(**A**) Occlusal view, (**B**) oblique view, and (**C**) lateral view of the planned endodontic access cavities (green cylinders) and access cavities performed using the augmented reality appliance (red cylinders).

**Figure 4 ijerph-19-11167-f004:**
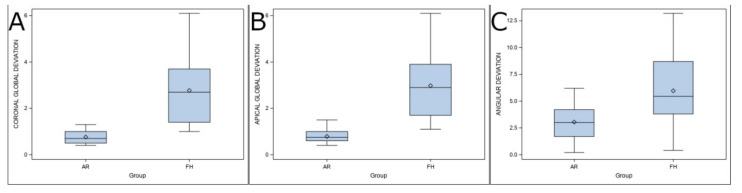
(**A**) Box plot of deviations observed in the study groups at the coronal entry point, (**B**) apical endpoint, and (**C**) angular deviations. The horizontal lines in each box represent the median values.

**Figure 5 ijerph-19-11167-f005:**
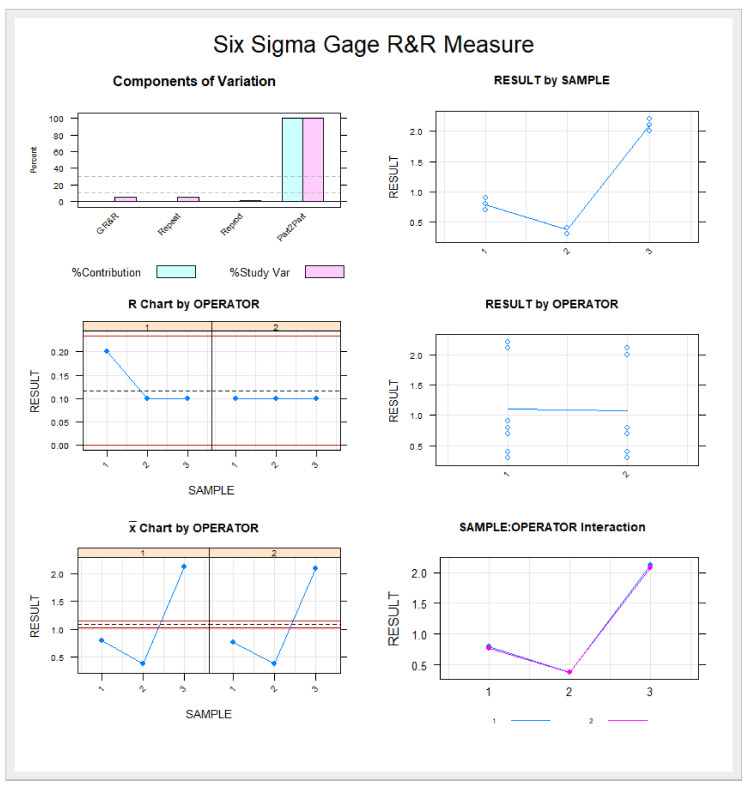
Measurement evaluation chart of the measurements technique indicating the difference between the measurements of each observer to evaluate the impact of each variable on the total variation obtained (components of variation) with a mean control chart and a range control chart (R chart by operator and x chart by appr), graphed measurement points (result by sample and result by operator), and interactions (sample–operator interaction). The values are within the confidence limits.

**Table 1 ijerph-19-11167-t001:** Coronal- (mm), apical- (mm), and angular- (°) level descriptive deviation levels and statistical significance (*p*-value).

	Group	*N*	Mean	SD	Minimum	Maximum	*p*-Value
Coronal	AR	30	0.76 ^b^	0.28	0.40	1.30	0.001
	MN	30	2.77 ^a^	1.35	1.00	6.10	
Apical	AR	30	0.79 ^b^	0.28	0.40	1.50	0.001
	MN	30	2.98 ^a^	1.41	1.10	6.10	
Angular	AR	30	3.05 ^b^	1.72	0.20	6.20	0.001
	MN	30	5.97 ^c^	3.42	0.40	13.00	

AR: augmented reality appliance; MN, manual navigation. ^a,b,c^ Statistically significant differences between groups (*p* < 0.05).

**Table 2 ijerph-19-11167-t002:** Descriptive statistics of the measurement variables.

Operator	*n*	Mean	SD	Minimum	Maximum
A	6	0.800	0.063	0.700	0.900
	6	0.383	0.041	0.300	0.400
	6	2.117	0.041	2.100	2.200
B	6	0.767	0.052	0.700	0.800
	6	0.383	0.041	0.300	0.400
	6	2.083	0.041	2.000	2.100

## Data Availability

Data are available upon request as per any relevant considerations (e.g., privacy or ethical restrictions).

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
