# Peer review of "Accuracy of Endodontic Access Cavities Performed Using an Augmented Reality Appliance: An In Vitro Study"

_ijerph, 2022, doi:10.3390/ijerph191811167_

Round 1
Reviewer 1 Report
Dear Authors, your research is well written and organized. Augmented reality is an innovative field in Endodontics and your study design and experimental procedure are well done. I have a few comments:
1-In my opinion the sample number could be increased.
2-can you explain why only 10 samples for each group?
3- In the introduction, after this phrase: "Bacteria are the main cause of periapical and pulpal inflammation, and failure to adequately eradicate bacteria and their by-products may lead to persistent irritation and impede the healing process [ 4 ]" I suggest you explain more about the importance of the irrigation phase in Endodontics and I suggest you add this reference:
Iandolo A, Simeone M, Orefice S, Rengo S. 3D cleaning, a perfected technique: thermal profile assessment of heated NaOClDetersione 3D, una tecnica perfezionata: valutazione dei profili termici dell’NaOCl riscaldato. Giornale Italiano di Endodonzia, 2017 31, 1, 58-61.
4- regarding the results and the conclusions they are well written.
Author Response
Dear Reviewer 1:
I’m pleased to resubmit the manuscript of the work entitled, “Accuracy of of Endodontic Access Cavities Performed Using an Augmented Reality Appliance: An in Vitro Study”
Reviewer 1: English language and style are fine/minor spell check required
Response: In order to adapt to the reviewer's 1 comments, we have sent the manuscript to the English Editing Service of MDPI. We attached the Certificate.
Reviewer 1: Dear Authors, your research is well written and organized. Augmented reality is an innovative field in Endodontics and your study design and experimental procedure are well done. I have a few comments: 1-In my opinion the sample number could be increased.
Response: In order to adapt to the reviewer's 1 comments, we have consulted a statistician about the sample size, and he replied that: “In this study, statistically significant differences between groups are observed, so the sample size is already large enough. When the differences between means are large and the deviations small, a sample size of 5-10 observations per group is sufficient”. However, the authors have increased the sample size up to 30 samples per group and we have consequently modified the results based on these changes.
Reviewer 1: 2-can you explain why only 10 samples for each group?
Response: In order to adapt to the reviewer's 1 comments, we have consulted a statistician about the sample size, and he replied that: “In this study, statistically significant differences between groups are observed, so the sample size is already large enough. When the differences between means are large and the deviations small, a sample size of 5-10 observations per group is sufficient”. However, the authors have increased the sample size up to 30 samples per group and we have consequently modified the results based on these changes.
Reviewer 1: 3 In the introduction, after this phrase: "Bacteria are the main cause of periapical and pulpal inflammation, and failure to adequately eradicate bacteria and their by-products may lead to persistent irritation and impede the healing process [ 4 ]" I suggest you explain more about the importance of the irrigation phase in Endodontics and I suggest you add this reference:
Iandolo A, Simeone M, Orefice S, Rengo S. 3D cleaning, a perfected technique: thermal profile assessment of heated NaOClDetersione 3D, una tecnica perfezionata: valutazione dei profili termici dell’NaOCl riscaldato. Giornale Italiano di Endodonzia, 2017 31, 1, 58-61.
Response: In order to adapt to the reviewer's 1 comments, we have added a sentence and the recommended reference in the Introduction section: “therefore, many authors have highlighted the relevance of the sodium hypochlorite disinfection agent to remove bacteria inside the root canal system; moreover, Iandolo et al reported that heating sodium hypochlorite increases the antimicrobial activity of the irrigating agent [5]”.
We take this opportunity to thank the recommendations and suggestions made by the reviewers to improve the document.
Yours sincerely,
Reviewer 2 Report
Dear authors!
This research is well performed and this work good both in terms of the quality of the presentation of the results and the content. I have no criticisms or contradictions. Asking authors questions, I expect those answers that just fit into the correction for Accept after minor revision (corrections to minor methodological errors and text editing). It would be useful for the authors to add more references to similar experiences of other researchers in the field of digital dentistry in the introduction. However, taking into account your letter, I will add that the authors should explain whether the results of the study are reliable given the small sample? Also The questions asked before remain relevant. 1. Did you perform x-ray/ct research before dividing into the groups? According to the data (periodontal disease) root canals can be obliterated which can also be a cause of changes in working protocols. 2. Please enlarge the conclusion part and give some recommendations to those colleagues who can use your technology in clinical practice
Author Response
Dear Reviewer 2,
I’m pleased to resubmit the manuscript of the work entitled, “Accuracy of of Endodontic Access Cavities Performed Using an Augmented Reality Appliance: An in Vitro Study”
Reviewer 2: English language and style are fine/minor spell check required
Response: In order to adapt to the reviewer's 2 comments, we have sent the manuscript to the English Editing Service of MDPI. We attached the Certificate.
Reviewer 2: This research is well performed and this work good both in terms of the quality of the presentation of the results and the content. I have no criticisms or contradictions. Asking authors questions, I expect those answers that just fit into the correction for Accept after minor revision (corrections to minor methodological errors and text editing). It would be useful for the authors to add more references to similar experiences of other researchers in the field of digital dentistry in the introduction.
Response: In order to adapt to the reviewer's 2 comments, we have added more references to similar experiences of other researchers in the field of digital dentistry in the introduction section: “Similar methodology has been previously used by Fraguas de San José et al to analyze the wear of dental implant drills after clinical using [Fraguas de San José L, Ruggeri FM, Rucco R, Zubizarreta-Macho Á, Alonso Pérez-Barquero J, Riad Deglow E, Hernández Montero S. Influence of Drilling Technique on the Radiographic, Thermographic, and Geomorphometric Effects of Dental Implant Drills and Osteotomy Site Preparations. J Clin Med. 2020, 9, 3631. doi: 10.3390/jcm9113631.], Alakabani et al to analyze the removal capability of carrier-based root canal filling material from straight root canal systems [Alakabani TF, Faus-Llácer V, Faus-Matoses I, Ruiz-Sánchez C, Zubizarreta-Macho Á, Sauro S, Faus-Matoses V. The Efficacy of Rotary, Reciprocating, and Combined Non-Surgical Endodontic Retreatment Techniques in Removing a Carrier-Based Root Canal Filling Material from Straight Root Canal Systems: A Micro-Computed Tomography Analysis. J Clin Med. 2020, 9, 1989. doi: 10.3390/jcm9061989.] and Faus-Matoses et al to analyze the wear of CM-Wire NiTi alloy endodontic reciprocating files after root canal treatment [Faus-Matoses V, Faus-Llácer V, Aldeguer Muñoz Á, Alonso Pérez-Barquero J, Faus-Matoses I, Ruiz-Sánchez C, Zubizarreta-Macho Á. A Novel Digital Technique to Analyze the Wear of CM-Wire NiTi Alloy Endodontic Reciprocating Files: An In Vitro Study. Int J Environ Res Public Health. 2022, 19, 3203. doi: 10.3390/ijerph19063203.]”.
Reviewer 2: However, taking into account your letter, I will add that the authors should explain whether the results of the study are reliable given the small sample? Also The questions asked before remain relevant.
Response: In order to adapt to the reviewer's 2 comments, we have consulted a statistician about the sample size, and he replied that: “In this study, statistically significant differences between groups are observed, so the sample size is already large enough. When the differences between means are large and the deviations small, a sample size of 5-10 observations per group is sufficient”. However, the authors have increased the sample size up to 30 samples per group and we have consequently modified the results based on these changes.
Reviewer 2: 1. Did you perform x-ray/ct research before dividing into the groups?
Response: In order to adapt to the reviewer's 2 comments, we clarify that a preoperative CBCT scan was performed after randomizing the teeth into the study groups; however, the samples were thoroughly selected according to the following inclusion criteria: “The teeth all presented without any caries or restorations, and they had been extracted for periodontal reasons. The clinical crown dimensions of the teeth ranged from 8.5–9.5 mm in height, 4.5–5.5 mm in M–D width, and 5.5–6.5 mm in V-L/P width”.
Reviewer 2: According to the data (periodontal disease) root canals can be obliterated which can also be a cause of changes in working protocols.
Response: In order to adapt to the reviewer's 2 comments, we clarify that the cariogenic process can induce the tertiary dentine formation which obliterate the dentin-pulp complex leading to a calcific metamorphosis; however, the periodontal disease does not use to activate the odontoblasts to induce the reactionary tertiary dentine formation. Moreover, the authors performed a preoperative CBCT scan after randomizing the teeth into the study groups to analyze the dentin-pulp complex space, discarding the teeth affected by calcific metamorphosis.
Reviewer 2: 2. Please enlarge the conclusion part and give some recommendations to those colleagues who can use your technology in clinical practice
Response: In order to adapt to the reviewer's 2 comments, we have modified the Conclusion section:” augmented reality technology has been shown as a viable alternative for access cavities in endodontics; however, clinical studies are required to validate this procedure clinically”.
We take this opportunity to thank the recommendations and suggestions made by the reviewers to improve the document.
Yours sincerely,
Reviewer 3 Report
The paper is based on an important aspect of dentistry, the article is written with a lot of passion, however, the following suggestions need to be included/clarify prior to publication.
Title: Check the title for typo errors
Methods:
1. The reliability and validity of the method and measurement should be included in the section.
2. how many operators, the examiner calibration was performed or not...if not why?
3. reliability and validity assessment are necessary to produce an unbiased outcome.
4. the AR equipment was calibrated before use? who did the calibration? the functionality was checked by a trained person? how many sessions in total were performed?
Results:
1. clarify whether student t-test or paired t-test was applied for AR and manual technique mean values.
2. student t-test is usually used against a known value
3. should include the p values and then mark with an asterisk for clarity.
Discussion:
1. expand the section with relevant studies like
Fahim S, Maqsood A, Das G, Ahmed N, Saquib S, Lal A, Khan AA, Alam MK. Augmented Reality and Virtual Reality in Dentistry: Highlights from the Current Research. Applied Sciences. 2022 Apr 7;12(8):3719.
2. Add limitations of the study, and mention future recommendations of the study.
3. include the strengths of the study.
conclusion:
repetition of the results section, write main achievements and how it is gonna help the field in your own words
Author Response
Dear Reviewer 3,
Reviewer 3: English language and style are fine/minor spell check required
Response: In order to adapt to the reviewer's 3 comments, we have sent the manuscript to the English Editing Service of MDPI. We attached the Certificate.
Reviewer 3: The paper is based on an important aspect of dentistry, the article is written with a lot of passion, however, the following suggestions need to be included/clarify prior to publication.: Title: Check the title for typo errors
Response: In order to adapt to the reviewer's 3 comments, we have revised the Title of the manuscript.
Reviewer 3: Methods: 1. The reliability and validity of the method and measurement should be included in the section.
Response: In order to adapt to the reviewer's 3 comments, we clarify that the repeatability and reproducibility of the digital measurement technique was analyzed using Gage R&R statistical analysis. The variability attributable to the measurement technique of the total variability of the samples was 2.7% for repeatability and 2% for reproducibility. The authors added this information in the Material and Methods and Results section.
Reviewer 3: 2. how many operators, the examiner calibration was performed or not...if not why?
Response: In order to adapt to the reviewer's 3 comments, we confirm that the access cavities were performed by a unique operator, except for the 6 access cavities made by a second operator to analyze the reproducibility of the measurement technique. The degree of agreement between the examiners was evaluated using the kappa index (Cohen J. A coefficient of agreement for nominal scales. Educ Psychol Meas 1960; 20: 37-46.). The proportion of correct answers observed ranged between 86% and 90% and the values of the kappa test between 0.61-0.69, values that are considered substantial based on the Landis and Koch scale (Landis JR, Koch GG. The measurement of observer agreement for categorical data. Biometrics 1977; 33(1):159-74.). The authors added this information in the Material and Methods section.
Reviewer 3: 3. reliability and validity assessment are necessary to produce an unbiased outcome.
Response: In order to adapt to the reviewer's 3 comments, we clarify that the repeatability and reproducibility of the digital measurement technique was analyzed using Gage R&R statistical analysis. The variability attributable to the measurement technique of the total variability of the samples was 2.7% for repeatability and 2% for reproducibility. The authors added this information in the Material and Methods and Results section.
Reviewer 3: 4. the AR equipment was calibrated before use? who did the calibration? the functionality was checked by a trained person? how many sessions in total were performed?
Response: In order to adapt to the reviewer's 3 comments, we clarify that the AR device was calibrated before use by a specialist with 5 years of experience in AR technology, who also checked the functionality of the AR device and trained the operator during 5 sessions of 5 hours each.
Reviewer 3: Results: 1. clarify whether student t-test or paired t-test was applied for AR and manual technique mean values.
Response: In order to adapt to the reviewer's 3 comments, we clarified in the Results section that we have used a two-sample t-test in the Results section.
Reviewer 3: 2. student t-test is usually used against a known value
Response: In order to adapt to the reviewer's 3 comments, we clarified in the Results section that we have used a two-sample t-test in the Results section.
Reviewer 3: 3. should include the p values and then mark with an asterisk for clarity.
Response: In order to adapt to the reviewer's 3 comments, we have added the p-values and highlighted with an asterisk in the Results section.
Reviewer 3: Discussion: 1. expand the section with relevant studies like
Fahim S, Maqsood A, Das G, Ahmed N, Saquib S, Lal A, Khan AA, Alam MK. Augmented Reality and Virtual Reality in Dentistry: Highlights from the Current Research. Applied Sciences. 2022 Apr 7;12(8):3719.
Response: In order to adapt to the reviewer's 3 comments, we have added a sentence and reference in the Discussion section.
Reviewer 3: 2. Add limitations of the study, and mention future recommendations of the study.
Response: In order to adapt to the reviewer's 3 comments, we have added the following sentence in the Discussion section: “The main strength of this study is the novel approach of augmented reality application for access cavities design. However, the experimental nature of this study avoid transferring the results sto the clinical setting; therefore, further clinical studies are required to validate this technique for endodontic application.
Reviewer 3: 3. include the strengths of the study.
Response: In order to adapt to the reviewer's 3 comments, we have added the following sentence in the Discussion section: “The main strength of this study is the novel approach of augmented reality application for access cavities design. However, the experimental nature of this study avoid transferring the results sto the clinical setting; therefore, further clinical studies are required to validate this technique for endodontic application.
”.
Reviewer 3: conclusion: repetition of the results section, write main achievements and how it is gonna help the field in your own words
Response: In order to adapt to the reviewer's 3 comments, we have modified the Conclusion section:” augmented reality technology has been shown as a viable alternative for access cavities in endodontics; however, clinical studies are required to validate this procedure clinically”.
We take this opportunity to thank the recommendations and suggestions made by the reviewers to improve the document.
Yours sincerely,
Round 2
Reviewer 3 Report
The paper has been amended as per changes requested.